# Oxidative Stress and BPA Toxicity: An Antioxidant Approach for Male and Female Reproductive Dysfunction

**DOI:** 10.3390/antiox9050405

**Published:** 2020-05-10

**Authors:** Rosaria Meli, Anna Monnolo, Chiara Annunziata, Claudio Pirozzi, Maria Carmela Ferrante

**Affiliations:** 1Department of Pharmacy, University of Naples Federico II, Via Domenico Montesano 49, 80131 Naples, Italy; meli@unina.it (R.M.); chiara.annunziata@unina.it (C.A.); 2Department of Veterinary Medicine and Animal Productions, Federico II University of Naples, Via Delpino 1, 80137 Naples, Italy; anna.monnolo@unina.it

**Keywords:** bisphenol A, oxidative stress, reproduction, gender toxicity, antioxidants

## Abstract

Bisphenol A (BPA) is a non-persistent anthropic and environmentally ubiquitous compound widely employed and detected in many consumer products and food items; thus, human exposure is prolonged. Over the last ten years, many studies have examined the underlying molecular mechanisms of BPA toxicity and revealed links among BPA-induced oxidative stress, male and female reproductive defects, and human disease. Because of its hormone-like feature, BPA shows tissue effects on specific hormone receptors in target cells, triggering noxious cellular responses associated with oxidative stress and inflammation. As a metabolic and endocrine disruptor, BPA impairs redox homeostasis via the increase of oxidative mediators and the reduction of antioxidant enzymes, causing mitochondrial dysfunction, alteration in cell signaling pathways, and induction of apoptosis. This review aims to examine the scenery of the current BPA literature on understanding how the induction of oxidative stress can be considered the “fil rouge” of BPA’s toxic mechanisms of action with pleiotropic outcomes on reproduction. Here, we focus on the protective effects of five classes of antioxidants—vitamins and co-factors, natural products (herbals and phytochemicals), melatonin, selenium, and methyl donors (used alone or in combination)—that have been found useful to counteract BPA toxicity in male and female reproductive functions.

## 1. Introduction

Endocrine-disrupting chemicals (EDCs) are a heterogeneous group of substances that are able to interfere with the hormonal-signaling pathways and alter metabolic and reproductive functions. Among EDCs, bisphenol A (BPA) is a non-persistent anthropic compound widely employed in the last decades as a component of plastic for multiple applications (e.g., food packaging, personal care, household products, and medical devices) so it is now environmentally ubiquitous. Indeed, BPA has been detected in the atmosphere, soil, and aquatic environments, as well as in foodstuff, house dust, consumer products, and biotic matrices [1,2].

Exposure to BPA is associated with deleterious health effects for animals and humans and affects not only endocrine and reproductive organs but also immune and central nervous systems through several mechanisms, including oxidative stress [3]. Growing evidence from research on laboratory animals shows that this non-persistent compound alters male and female reproductive function even at extremely low exposure levels [4]. This is relevant because BPA exposure may be chronic, making it functionally equivalent to a persistent compound.

The mechanisms underlying BPA toxicity may be related to its chemical properties in the body, including interactions with hormone receptors (i.e., estrogen receptors) in target cells [5]. As a weak estrogen receptor ligand, BPA binds to the classical nuclear receptors with different affinity, for example, estrogen receptor (ER)α less than ERβ (about 10,000 times lower than 17-β-estradiol), and its potency of action becomes higher when the estradiol level is at very low concentrations [6]. BPA also binds many other membranes and nuclear receptors, activating them at concentrations lower than those required to activate ERs [7]. In particular, BPA binds to the membrane-bound form of estrogen receptor (mER)α and G protein-coupled receptor (GPR) 30, inducing non-genomic effects. GPR30 has been detected in several cell types, and it is involved in cell proliferation and apoptosis. Indeed, seminoma proliferation is induced by a low concentration of BPA through a GPR30 non-genomic activation [8]. BPA also binds to another estrogen-related receptor (ERR)γ, an orphan nuclear receptor highly expressed in the fetal brain and placenta by which BPA influences fetal tissue development [9].

Widespread contamination and dietary ingestion have led to BPA exposure in the general population, as evidenced by the scientific literature on BPA occurrence in human tissues and body fluids (i.e., urine, serum, plasma, saliva, breast milk, semen, follicular fluids, and adipose tissues) [10].

A recent epidemiological study has hypothesized that higher urinary BPA concentrations in humans might negatively impact female reproduction [11]. Moreover, in men, urinary BPA levels have been correlated to abnormal semen parameters [12], and BPA-exposed individuals also showed reduced libido and erectile ejaculatory difficulties. The overall BPA effects on male reproduction appear to be more harmful if exposure occurs in utero [7,13].

Given the challenges in both human exposure assessment and tissue sample collection, animal models are important for assessing the underlying toxic mechanisms observed in humans. First, differently from humans, rodents allow histological evaluation of various tissues, as well as for tissue- and cellular-specific molecular analysis. Second, rodent models provide excellent opportunities for sex-specific evaluations. In the experimental design, important factors must be considered, including the examined tissue, the structural, functional, and epigenetic differences between gender and species, the exposure time, and chemical dose. Third, despite the limitations of clinical studies, many animal studies have evidenced that BPA affects several reproductive endpoints.

BPA toxicity is associated with oxidative stress and related markers in several experimental models [3] and humans [14]. Mounting evidence shows that the production of reactive oxygen species (ROS) and/or decreased capacity of antioxidant defense significantly contribute to BPA organ toxicity, altering the oxidative balance in the mitochondria and generally in the cell [3,15].

This review offers new insights and up-to-date literature survey on these issues to identify oxidative stress as the leitmotif underlying BPA’s mechanism of action with pleiotropic outcomes on reproduction. Besides, the review also examines the effect of antioxidant substances on the BPA-induced toxicity and analyzes their possible efficacy as a therapeutic strategy to limit the damage on female and male reproductive organs and their functions.

## 2. Oxidative Stress as a Mark of BPA Toxicity

Oxidative stress is a key component of inflammatory reactions and has been implicated in aging, cardio-metabolic, and immune diseases, neuronal degeneration, and the development and progression of cancer, with different and not completely understood mechanisms [16]. The regulation of redox balance is essential to maintain cellular homeostasis, development growth, and survival. Indeed, physiological cellular metabolism generates ROS (i.e., superoxide anions, peroxides, and hydroxyl radicals) involved in redox balance, a well-orchestrated process. ROS comprise not only radical and nonradical oxygen derivatives but also nitrogen-containing compounds defined as reactive nitrogen species (i.e., nitric oxide and peroxynitrite). However, few data have related BPA toxicity to nitrosative species, in particular in the reproductive system. Redox balance is coordinated by numerous cellular components to avoid excess ROS and to prevent related deleterious effects (mutations, unchecked cell growth, and insensitivity to cell death signals).

As a metabolic and endocrine-disrupting chemical, BPA can impair oxidative homeostasis via direct or indirect mechanisms, including the increase of oxidative mediators and reduction of antioxidant enzymes, determining mitochondrial dysfunction, alteration in cell signaling pathways, and induction of apoptosis [17,18]. BPA induces oxidative stress by decreasing antioxidant enzymes (superoxide dismutase (SOD), catalase, glutathione reductase (GR), and glutathione peroxidase (GSH-Px)) and increasing hydrogen peroxide and lipid peroxidation in the liver and epididymal sperm of rats; it can also alter organogenesis of the kidney, brain, and testis in mice [19,20,21]. 

The ROS increase induced by BPA has been reported in several cell types with concentrations ranging from nanomolar to micromolar [3,22]. Differences in ROS generation, duration, and cytotoxicity appear mainly based on the concentration of BPA, on cellular backgrounds, exposure time, and sensor fluorescent reagents [3,23].

BPA has been recently associated with inflammation markers in addition to oxidative stress in humans [24]. Previously, Yang et al. [14] conducted a cross-sectional study on men and premenopausal/postmenopausal women; the authors observed that urinary BPA levels were positively associated to urinary oxidative stress and blood inflammatory biomarkers (alteration in white blood cell count and C-reactive protein) in postmenopausal women, evidencing that BPA effects depend on the gender and hormonal status. These authors assumed that the gender-related effect of BPA toxicity might be due to the estrogen level and receptor occupancy. As is known, the ER expression is gender and age-related [25]. In postmenopausal women, the hormone level is very low, allowing a more extensive binding of xenoestrogen BPA to ER, triggering noxious cellular responses associated with oxidative stress and inflammation. 

A strong relationship between oxidative stress and inflammation has been documented; these reactions are inextricably linked to cellular processes, and they may be both causes and/or consequences of cell damage. The ROS and nitrogen species generated by immune cells, as macrophages and neutrophils, contribute to the establishment of chronic inflammation. Malondialdehyde (MDA), an indicator of lipid peroxidation, and 8-hydroxy-deoxyguanosine (8-OHdG), a marker of DNA oxidation, have been widely considered as potential biomarkers for oxidative stress [26,27] and several reports show their increase in BPA-exposed organisms [28]. 

The ROS-induced DNA base modification has been shown to induce the nuclear factor kappa-light-chain-enhancer of activated B cells (NF-κB) pathway of inflammation. Many other redox-sensitive signal transduction pathways like c-Jun N-terminal kinase (JNK) and p38 mitogen-activated protein (MAP) kinase and transcription factor AP-1 also participate and sustain the vicious cycle between inflammation and oxidative stress [29].

The oxidative stress induced by BPA has been associated with human B-cells cytotoxicity [30] and to the impairment of immune response in vitro in murine macrophages [31], as well as in animal models [32]. BPA exposure also exacerbates the expression of proinflammatory cytokines, as well as interleukin (IL)-1β, IL-6, IL-8, and tumor necrosis factor-α (TNF-α), in many tissues and organs (i.e., serum, colon, liver) [17] and the hydrogen peroxide species and ROS induced by BPA activates NF-κB [22]. Furthermore, BPA-induced oxidative stress plays an important role in the activation of NOD-like receptor protein 3 (NLRP3) inflammasome with inflammatory co-stimulus (i.e., nitric oxide) in adipocyte cells [33] or the liver of BPA-exposed obese mice (our unpublished results, presented as oral communication to the international congress of the British Pharmacological Society (2019), Edinburgh, UK).

Behind lipid peroxidation and inflammation reactions, the mechanisms of oxidative stress induced by BPA also include mitochondrial dysfunction. Low doses of BPA impair mitochondrial function in the liver [15], leading to hepatic toxicity. Moreover, this pollutant induces ATP depletion, the release of cytochrome c, loss of mitochondrial mass and membrane potential, and alterations in the expression of genes involved in mitochondrial activity and metabolism [34].

As is known, the excess of oxidative stress is involved in the neurodegeneration of many brain disorders [35]. Tavakkoli et al. [18] evidenced the neurotoxicity of BPA on the alteration of the expression of functional proteins influenced by the oxidative status that is related to neurological disorders as well as schizophrenia, depression, epilepsy, and brain tumors.

Evidence suggests a gender difference in responsiveness to oxidative stress [36]. Females are less vulnerable to the protective effects of estrogen related to the hormone’s ability to act as an electron-donating antioxidant [37]. Moreover, in females, mitochondria have higher levels of reduced glutathione than those of males, and the oxidative damage to mitochondrial DNA is lower in females than in males, due to higher expression and activities of manganese SOD and GSH-Px [36,38]. Sex differences in oxidative stress depend not only on the higher antioxidant defense in females but also on the increased ROS generation in males [39]. Indeed, some studies have been reported a gender-dependent NADPH oxidase activity, resulting in the excessive formation of reactive intermediates in male mice [40,41]. Based on the intriguing literature concerning this issue, it will be relevant to carry out comparative studies on gender-related effects of BPA on oxidative damage.

## 3. Mechanisms Linking BPA-Induced Oxidative Stress to Reproductive Toxicity

### 3.1. BPA Genetic Damages

Chromosomal aberrations, including aneuploidy, DNA strand breaks, DNA adducts, and gene mutation, are the main genetic DNA damages. Although there has been a ubiquitous presence of BPA and a continuous co-exposure of humans with other pollutants, no data have been provided about direct BPA-induced DNA damage through the induction of oxidative stress and its deriving effects on reproduction.

In vitro and in vivo preclinical studies have shown that BPA causes DNA adduct formation [42], aneuploidy [43], and mutations [44]. These alterations may contribute to the onset of infertility [45], miscarriages [46], and birth defects [47]. Moreover, BPA suppresses the DNA repair pathway of base excision induced by laser irradiation [48]. 

BPA-induced DNA damage has been related to oxidative stress through the MDA, phenoxyl radicals, and ROS production [49,50], which can induce direct DNA alterations amplified by the impairment of antioxidant enzyme pathways [51]. Anet et al. [52] observed increased ROS production and lipid peroxidation when *Drosophila melanogaster* was exposed to BPA, by food, evidencing decreased antioxidant function and mutagenic effects. Moreover, Guo et al. [53] showed that treatment with BPA on porcine embryos increased ROS formation, causing cytochrome c release, mitochondrial and DNA damage, and apoptosis through p53-p21 signaling pathways. Following ex vivo exposure of rat ovaries to BPA, Ganesan et al. [54] showed DNA damage also sustained by BPA alkylating metabolite. 

Several human studies reported the correlation between BPA urinary levels, MDA, and/or 8-OHdG [45,55]. Guida et al. [56] revealed a positive correlation between BPA blood levels and chromosomal and fetal malformations in women with a diagnosis of developmental defects without a clear correlation with ROS production.

The BPA genotoxic effect was also observed in prostate epithelial cells: after 24h treatment with increasing concentrations, BPA downregulated the expression of p53 [57]. De Flora et al. [58] observed the presence of DNA adducts in prostate cells treated with high or low concentrations of BPA; the same authors exposed male rats to BPA (200 mg /kg for 10 days) evidencing a significant increase of MDA and nuclear spreading factor in spermatozoa, suggesting the fragility of these germ cells.

Micromolar BPA concentrations were reported to induce ROS formation, DNA damage, and cytotoxicity in spermatogonia (GC-1 cell line) [59]. Interestingly, the induction of DNA strand breaks and its correlation with ROS production were also observed at non-cytotoxic micromolar and nanomolar concentrations [48,50]. However, the induction of strand breaks by non-cytotoxic concentrations of BPA was not observed in other cell lines [60,61]. Some authors have evidenced that exposure to antioxidant substances reduces strand break signaling [50]. 

The contrasting opinions about the genotoxic effects of BPA on DNA may be related to differences in concentrations of BPA used, often higher than those measured in environmental matrices [60,61]. BPA induces genotoxic effects in vitro in somatic and germ cells, but the lack of carcinogen effect in animals does not allow us to consider BPA as a “real” genotoxic agent in vivo [62]. Being unable to directly cause genetic damage at environmental concentrations, it remains to be elucidated if BPA may exert an indirect genotoxic activity in the presence of other environmental pollutants or pathological conditions.

### 3.2. BPA Epigenetic Effects 

Epigenetic mechanisms manage gene expression and cellular phenotypes without modifying the nucleotide sequence of the genes, with different patterns that vary depending on cell type. Several studies have shown that DNA impairment and epigenetic modifications adversely impact early embryonic development [63]. During this period, extensive programming and reprogramming of DNA methylation and histone modification patterns occur, and the epigenetic mechanisms can modulate male and female genomes in a tissue-specific manner by activation/deactivation processes [64]. Methylation provokes gene silencing by proteins responsible for gene repression or inhibits the binding of transcription factors to DNA playing a pivotal role in mammalian development [65]. Literature data report the epigenetic effects of BPA, among other environmental pollutants, on reproduction depending on exposure mode (dose, time, and route), sex, and life stage [66]. 

In female agouti yellow mice, the administration of BPA (50 mg/kg in the diet from the mating to the weaning stage) induces a DNA hypomethylation causing phenotypic changes in the coat color of offspring [67]. Conversely, Doshi et al. [68] observed that, during the first five days of life, BPA treatment-induced hypermethylation of estrogen receptor promoter areas in newborn rat testicles, leading to the disruption of spermatogenesis process and fertility. BPA exposure, in utero and during neonatal development, may determine both DNA hypermethylation and hypomethylation in CpG islands in a dose-dependent manner, starting from low doses [69]. Other controversial data report opposite effects on DNA methylation due to BPA-induced epigenetic modifications and the underlying molecular and cellular mechanisms remain to be elucidated. Indeed, elevated ROS production and oxidative stress have been associated with chromosomal degradation with both DNA hypermethylation and hypomethylation [70].

Menezo et al. [71] determined that BPA induces epigenetic reproductive damage, and sperm epimutations and oxidative stress might be epigenetically inherited by offspring via DNA methylation. When pregnant albino rats were orally exposed to BPA (during gestation and/or lactation periods), it was observed an increase in MDA levels and DNA hypermethylation at the CpG islands in both DNA methyltransferases and ERα promoter genes in male offspring testis [72]. The decrease in superoxide dismutase (SOD), GST, and GSH-Px levels was also described. This evidence is consistent with our unpublished results on liver metabolic damage induced by BPA in obese mice. In our experimental condition, 21-day BPA oral exposure (50 μg/kg daily in corn oil) exacerbates the global DNA hypermethylation (% of methyl cytosine/total) induced by high-fat feeding (our unpublished results, presented as oral communication to the international congress of the British Pharmacological Society (2019), Edinburgh, UK).

An in vitro study reported that acute BPA increased ROS production, DNA methylation, and apoptosis/autophagy rates, decreased maturation rates, and induced abnormal cytoskeletons in porcine oocytes [73]. The authors concluded that BPA may further interfere through the impaired DNA methylation status in the meiotic progression of oocytes. 

Guo et al. [53] exposed porcine embryonic-derived parthenotes to BPA; the increase of ROS production was associated with mitochondrial and DNA damage leading to autophagy and alteration of DNA methylation (decreasing 5-methyl-cytosine and DNA methyltransferase mRNA), with a possible effect on offspring health. Conversely, Yuan et al. [74] studied the effects of a subacute BPA exposure on the testes of adult male minnow *Gobiocypris rarus*, evidencing the increase of hydrogen peroxide production and hypermethylation of global DNA. Very recently, Fan et al. [75] exposed in vitro grass carp ovary cells to BPA for 48 h, and observed a dose-dependent increase of ROS and MDA levels. Interestingly, the increase of global 5-methyl-cytosine levels was characterized by an inverted U-shaped trend, showing a discrepancy in the dose-response BPA behavior. 

The aforementioned results highlight the crosstalk between oxidative stress and DNA methylation, evidencing a strong variability among cell type, time, and concentration exposure to BPA.

### 3.3. Endocrine and Metabolic Disruption

Endocrine disruptors can affect hormonal balance via alteration of hormone secretion/metabolism and binding hormone-receptor; all these activities result in a broad variety of developmental and reproductive abnormalities. 

Recent and different studies (in vitro, in vivo, and epidemiological ones) have linked EDC exposure with the increase of susceptibility to several pathologies such as obesity, metabolic syndrome, type 2 diabetes, and cardiovascular diseases. Indeed, some disruptors cause metabolic diseases per se, while others act increasing the sensitivity or the set point for the disease. Many difficulties have been evidenced to fairly assess the risks related to EDC exposure, depending on developmental, timing, and dose exposure, but also to the diversity of chemical structure and mechanisms of action.

BPA, as xenoestrogen, mimics the structure and function of 17-β estradiol [76]; it also binds classical and non-classical membrane estrogen receptors, including GPR30 [77]. In perinatally BPA-exposed females, these actions induce several changes in estrogen-target organs, including ovary, mammary gland, and uterus [76]. Moreover, BPA affects the transcription of target genes mediated by ERβ, inhibiting receptor degradation and ubiquitination [78].

Furthermore, BPA binds the androgen receptor, acting as an antagonist [79]. As is known, the androgen receptor is the major regulatory element of androgen cell signaling, and it is essential for male reproductive function and development.

The hypothalamic–pituitary–testicular axis and the activity of thyroid hormones coordinate human and animal spermatogenesis. In a human study, Liang et al. [80] showed the association between BPA exposure and increased serum levels of follicle-stimulating hormone (FSH) and luteinizing hormone (LH) and decreased levels of testosterone in Chinese men. As is known, FSH and LH increase testicular growth, acting on Sertoli and Leydig cells, respectively, and enhance testosterone synthesis. 

The adverse effect of BPA on male reproductive hormones have been previously reviewed, reporting similar effects on thyroid hormones [13] that are recognized to affect male fertility and sperm production. Regarding thyroid hormone levels, prenatal BPA exposure later in gestation reduced TSH in newborn girls [81], and ROS production by BPA may be involved in thyroid disease [82]. 

The relationship between BPA and metabolic diseases (i.e., obesity and diabetes) can affect male and female fertility and reproduction, increasing fat accumulation and inducing the impairment of glucose and lipid metabolism associated with inflammation. As a lipophilic substance, BPA accumulates in fat, increasing adipocyte number and size and contributing to weight gain [83]. As previously reviewed [84], BPA at environmental-occurring doses modulates adipokine release, reducing anti-inflammatory adiponectin and stimulating proinflammatory leptin and other inflammatory cytokines (i.e., IL-6, TNF-α). As is known, adipokines such as leptin and resistin and gut peptides are considered to be crucial in the interaction between energy balance and reproduction [85]. High levels of leptin are related to fat mass and leptin resistance that, similar to insulin-resistance, are common in obesity and related pathologies such as diabetes [86]. Interestingly, humans with higher BPA serum levels are more than twice prone to develop diabetes than those with lower BPA levels [87], with increased diabetes risk in lifespan timing [88]. 

### 3.4. Cell Signaling

BPA activates different and multiple cell signaling pathways depending on cell type and function. In non-genomic signaling, BPA binds GPR30 and mER and induces the activation of these receptors, evoking rapid estrogenic signaling activating kinase systems (i.e., protein kinase-A (PKA), MAPK, phosphoinositide 3-kinases (PI3K)) and modulating cAMP, protein kinase-C (PKC) and intracellular calcium levels in ovarian tumor cells [89]. It has been recently summarized that exposure of spermatozoa to BPA in vitro induces the activation of these receptors associate with increased ROS [90]. At low levels, ROS are important for normal sperm function, but high levels of ROS induce detrimental effects on mobility and acrosome reaction, loss of sperm motility, ionic imbalance and alteration of the sperm proteome [79], damaging cellular lipids, proteins and DNA. These alterations rapidly increase the phosphorylation of MAPK, PI3K, and PKA and elevated ROS availability in the cytoplasm. The increased ROS production causes phosphorylation of sperm proteins (in tyrosine residues), triggering functional modification of coregulatory protein/genes via transcriptional and translational mechanisms, and predisposing to abnormal cellular outcomes [91]. Thus, BPA can compromise the complex oxidative stress response in spermatozoa, affecting their fertilizing ability [90]. Increased ROS and lipid peroxidation by BPA have also been demonstrated in other cell types [3]. 

It had been reviewed that BPA affects many cell-signaling mechanisms and pathways (i.e., NF-κB, extracellular signal-regulated kinases (ERK), JNK) involved in inflammation, immune response, cancer, brain, and reproductive function; the increase of calcium and ROS and the activation of ERK and JNK are implicated in BPA-induced apoptotic cell death [92]. 

Several studies have determined that EDC-induced oxidative stress can impair male fertility by disrupting the cell junctions among cell testis. Wong and Cheng [93] reviewed the pathways involved in BPA male reproductive toxicity, analyzing the pathophysiological role of MAPK and inflammatory cytokines and that of the PI3K/c-Src/ focal adhesion kinase signaling pathway. The resulting alteration of intercellular junctions, via polarity proteins, leads to reproductive dysfunction such as reduced sperm count and reduced semen quality [93].

## 4. Oxidative Stress-Induced BPA Toxicity on Male and Female Reproduction

Between 2000 and 2010, a lot of contradictory literature data on the effect of BPA on male fertility were collected, but many other recent reports have consistently established the adverse effects of BPA on male and female reproductive functions [13,94,95].

In particular, in males, BPA impairs spermatogenesis and sperm quality, as evidenced by trans-generational studies; in females, it alters ovary, embryo development, and egg cell quality [94]. Moreover, BPA impairs the reproductive function in offspring due to its capability to cross the blood–brain barrier [96], in addition to its direct effects on reproductive organs.

The alterations of reproductive functions have irreversible consequences on adult fertility, mainly if these damages occur during the development of the reproductive organ in fetal life [97]. In particular, oxidative stress induced by several EDCs, including BPA, is linked to male infertility [98]. Many preclinical findings have shown the impact of prenatal exposure of BPA on male spermatogenesis, via several mechanisms of action. Quan et al. [99] demonstrated that oral BPA exposure in dams (gestational days (GD)) [14,15,16,17,18,19,20,21] provokes an increased ROS production, the activation of the apoptotic pathway in the testis of male offspring, confirmed by histopathological changes. BPA exposure in CD-1 dams (GD 7-14) also led to the alteration of sperm quality and motility in adult male mice, with changes in oxidative balance [100]. Furthermore, in a murine model of chronic BPA exposure of dams, during all gestational period, and continuing the oral administration to male pups until to sexual maturation, damaged spermatogenesis was observed in progenies [101]. This impairment was associated with decreased antioxidant defense and reduced expression of sirtuin 1, a key sensor of ROS production.

Very recently, data have confirmed that male pups exposed early to BPA produced oxidative stress in the testicular tissue [72]. In this comparative study (among prenatal, perinatal, and postnatal BPA exposure), it was demonstrated that BPA-related epigenetic modifications occur independently from the time-exposure. Nevertheless, the magnitude of the BPA toxic effect on oxidative balance was time-exposure dependent, reaching a maximum effect during perinatal exposure (from pregnancy to lactation) [72].

BPA exposure in adult rats increased ROS amount in a dose-dependent manner and oxidative stress was associated to metabolic impairment (hyperglycemia and hyperinsulinemia) and reproductive damage, as demonstrated by testicular reduction of insulin receptor substrate-2 and glucose transporter-8, two key proteins involved in testicular energy metabolism and spermatogenesis [102]. In epididymis, testis and immune cells (i.e., lymphocytes and bone marrow) of adult rats, BPA provoked the significant increase of lipid peroxidation [103] that is correlated to the impairment of sperm quality [104] and the reduction of several antioxidant enzymes such as SOD, catalase, and non-enzymatic reduced glutathione [103]. Recently, Khalaf et al. [105] also reported an increased level of H_2_O_2_, lipid peroxidation, and depletion of the antioxidant defense systems in the testis of offspring from dams exposed to BPA. At the early stage of puberty, BPA impaired sperm production and gonadotropin secretions and altered seminiferous tubule epithelium morphology, increasing ROS production and reducing the antioxidant activity of catalase and SOD enzymes [106]. 

In testis, maintained alteration of the redox balance leads to ROS overproduction and the impairment of the antioxidant defense, resulting in increased oxidative stress that is the major cause of sperm dysfunction and male infertility [98]. It is important to clarify that the clinical studies clearly addressed BPA reproductive toxicity to oxidative stress [107]. However, considering the importance of oxidative balance in sperm physiology, oxidative stress could be the main detrimental mechanism underpinning BPA toxicity on the male reproductive system.

In seminiferous tubules, BPA reduced the activity of antioxidant enzymes and increased that of myeloperoxidase (MPO), substantiating inflammation and testicular dysfunction linked to high ROS production due to the peroxidation of spermatogenic cell membranes. These concurrent adverse effects negatively impact the mitotic and meiotic process of cells during spermatogenesis (i.e., spermatogonia, spermatocytes, spermatids, and spermatozoa) that result in an impaired quality and quantity of spermatozoa within seminiferous tubules.

Clinical findings have confirmed preclinical data: higher concentrations of BPA in the maternal placenta were associated with an increased risk of urogenital complications in children (i.e., cryptorchidism and hypospadias) [108]. A cohort study showed a positive correlation between BPA maternal exposure and oxidative stress markers in the blood sample of neonates [109]; notably, oxidative stress is a crucial driver of pregnancy complications [110]. 

Moreover, a relationship between high blood/urinary BPA levels and anomalous semen parameters in men occupationally exposed has been reported, which also showed reduced libido and ejaculatory and erectile dysfunction (reviewed in Manfo et al. [13]). Accordingly, Wang et al. [17] found a positive correlation between BPA concentrations and oxidative markers in the urinary sample of adult men collected from days 0 to 90, a time window consistent to that of the spermatogenesis process.

A growing line of evidence indicates that perturbations induced by BPA and its substitutes can also affect female fertility leading to the alteration of oocyte meiotic maturation [111]. Specifically, BPA has detrimental effects on oocyte progression and quality, chromosome segregation, increasing oxidative damage, along with oocyte apoptosis demonstrated both in vitro and in vivo studies [73,112]. Indeed, oxidative stress, as well as DNA and epigenetic modifications, plays a crucial role in oocyte maturation and interferes with apoptosis, contributing to the alteration of germ cell nest breakdown and reproductive function [113]. BPA can impair fertility, disrupting the physiological germ cell nest breakdown, as shown by the multiple transgenerational toxicities after in utero exposure in mice [114]. This process is mainly controlled by the decrease of estrogens at the birth of pups and is also linked to the onset of apoptosis [115]. During this critical stage, the exposure at different doses of BPA in mice blocks the natural apoptosis required for releasing oocyte from their germ cell nests [116] and induces an alteration of key ovarian genes involved in the regulation of oxidative stress [114].

Any error in oocyte maturation triggers other injuries, including infertility, pregnancy loss, and birth defects, and BPA negatively impacts all these disorders of female reproductive physiology by epigenetic mechanisms [117].

Gupta et al. [118] reported that BPA decreased the amplitude and frequency of spontaneous uterine contractions in isolated uterus, showing the estrous phase by involving nitric oxide release and nitrergic mechanisms. Moreover, BPA exposure in female rats induced the inhibition of steroidogenic enzymes, decreased level of estrogen, and eNOS overexpression in the ovary [119].

Some epidemiological studies show that EDCs exposure can affect pregnant women [120,121]; particularly, in placental mammals, BPA causes an unsuccessful pregnancy and the alteration of several placental molecular endpoints [23], including hormone-related mRNA expression, micro-RNA expression, and DNA methylation [122]. Other studies highlight the sex-specific relationship between BPA exposure and placental outcomes (reviewed in Strakovsky and Schantz [23]) because of the double origin and development of the placenta (maternal and fetal tissues) and the following sexually dimorphic responses to ED [123]. Consistently, it has been demonstrated that in CD-1 mice, daily exposure to BPA led to the reduction of placental mRNA expression of nuclear receptors disrupting the physiological function of the placenta [124]. Very recently, Song et al. [125] showed that in sheep, the exposure to BPA in the gestational period causes low birth weight mainly due to the placental dysfunction that differs between early and mid-gestation with stage-specific epigenetic alterations. In particular, BPA induces oxidative stress (i.e., nitrotyrosine formation) and inflammation (IL-1β), as well as lipotoxicity and modification of placental steroid milieu, in the later gestational period. 

BPA toxicity on female reproduction also involves the occurrence of endometriosis, miscarriage, and abnormalities of uterus morphology [94]. The common thread among these disorders is the increased oxidative stress [126], along with the activation of inflammatory signaling. Indeed, Cho et al. [22] have demonstrated the capability of BPA to induce oxidative stress and inflammatory mediators in human endometrial cells. Interestingly, these authors found a decrease in antioxidant enzymes in endometrial cells after BPA exposure, probably due to the excessive production of ROS related to BPA metabolization that exceeds the intracellular antioxidant defense capacity. Moreover, pregnant BALB/c mice exposed to different BPA doses revealed profound changes in endometrium morphology, characterized by hypertrophy and the formation of endometriosis-like structures in comparison to the not-treated mice [127].

## 5. Protective Effects of Antioxidants on BPA Toxicity in Reproductive Systems

Antioxidants are molecules of different origins that prevent oxidative stress in cells and tissues, inhibiting oxidation or overproduction of ROS. In the last decades, many reports have demonstrated the prevention/reduction of BPA-induced toxicity using several antioxidants such as vitamins C and E, N-acetylcysteine, lipoic acid, ginger extract, and gallic acid in several organs [128,129,130,131,132]. The beneficial effects associated with the supplementation of antioxidant agents involve the protection of quality, vitality, motility, and morphology of spermatozoa [133]. Interestingly, the antioxidants reduce oxidative stress in spermatozoa during cryopreservation and exert protective effects on sperm motility and fertilization potential [98]. Here, we examine different types of antioxidants, used alone or in combination, that have been found useful to counteract BPA toxicity in male and female reproductive systems. All these substances and their descriptions are summarized in Table 1.

### 5.1. Vitamins and Co-Factors

The effect of glutathione, vitamin C, and vitamin E on sperm function has been demonstrated [135,151,152]. In particular, these three compounds prevented motility loss and abnormal acrosome reaction induced in BPA-exposed spermatozoa, downregulating the excessive release of ROS and increasing intracellular ATP production. Increased ROS and their metabolites are mainly responsible for the decline in motility and fertilizing competence, acting on enzymatic systems of spermatozoa and on DNA, lipids, and proteins, causing an irreversible injury to cells [153]. Glutathione (5 mM) and vitamin E (2 mM) counteracted the compromised fertilization and early embryo development caused by BPA (100 μM) [154]. Glutathione is one of the major endogenous antioxidants, whose recycle depends on vitamin C and E levels, and vice versa [155]. Indeed, beyond its scavenging activity, ascorbic acid or vitamin C is also an electron donor in several reactions, and its cellular recycle occurs at the expense of NADH or glutathione. The term vitamin E includes a series of lipophilic and essential micronutrients, with cytoprotective and antioxidant effects [156].

The combination of vitamin C (50 mg/kg) and E (50 mg/kg), but not these vitamins alone, was also able to exert a protective effect in preventing apoptotic cell death in the ovaries of rats exposed to a high dose of BPA [134]. As known, the high level of vitamin E is detected in male germ cells [157], and it plays a key role in managing the lipid peroxidation occurring in testicular microsomes and mitochondria [158]. Evidence confirmed that vitamin E (4 mg/100 g bw) improved male fertility, protecting testicular cells and epididymal sperm from the apoptosis induced by BPA exposure in Wistar rats [135]. 

It has been recently demonstrated that in vitro 1,25-dihydroxyvitamin D_3_ (1,25D_3_; 0.1 μM) modulates the toxic effect of BPA (10 μM) on oxidative stress and particularly on mitochondrial function and dynamics in ovarian granulosa cells [136]. Moreover, 1,25D_3_ prevented mitochondrial DNA deletion induced by BPA in the female ovary [136]. Interestingly, serum vitamin D levels are found to be reduced in women strictly related to BPA exposure [159]. 

Among other vitamins, coenzyme Q10 (CoQ10) (100 μg/mL) was able to restore fertility, rescuing the reproduction toxicity associated with BPA exposure (500 μM) in the *Caenorhabditis elegans* germline [137]. CoQ10 partially counteracted BPA-induced DNA damage reducing oxidative stress through the scavenging of ROS and free radicals, restoring mitochondrial function and decreasing meiotic DNA double-strand breaks. Contextually, CoQ10 supplementation led to a reduction of aneuploid embryos levels and of chromosome defects in oocytes at diakinesis of BPA-treated worms [137].

α-Lipoic acid (LA) is a biological thiol present in all types of prokaryotic and eukaryotic cells that reduces oxidative stress, increasing the levels of other antioxidants and playing an essential role in mitochondrial damage [160]. LA is a biofactor of energy metabolism and greatly reacts toward free radicals and reduced oxidative stress, counteracting the decrease of glutathione levels, limiting the formation of lipid peroxides (LPOs), and restoring the physiological antioxidant enzyme balance [130]. El-Beshbishy et al. [138] reported that LA (20 mg/kg per os) reduced testicular and mitochondrial oxidative stress caused by chronic administration of BPA (10 mg/kg) in male rats. Specifically, LA normalized BPA-altered activity of key mitochondrial enzymes (i.e., succinate dehydrogenase, malate dehydrogenase, isocitrate dehydrogenase, monoamine oxidase, and NADH dehydrogenase) and increased enzymatic and non-enzymatic antioxidants of mitochondria, including glutathione reductase and peroxidase, SOD and catalase. Moreover, BPA detrimental activity on testicular acid phosphatase, alkaline phosphatase, and lactate dehydrogenase, as well as total testicular proteins, was also mitigated by LA supplementation [138]. 

Finally, in female Wistar rats, long-term treatment with oral LA (100 mg/kg), during BPA exposure (25 mg/kg, per os) for 30 days led to the reduction of ovarian oxidative damage, preventing lipid peroxidation and nitrosative stress, and even more in combination with vitamin E (20 mg/kg, per os) [139].

Evidence suggests that vitamin A (not a popular antioxidant) could play a role in protecting against oxidative stress damage. In particular, retinoic acid, which is a metabolite of vitamin A, was reported to upregulate the expression of anti-oxidant related genes in in-vitro mature buffalo oocytes [161]. Moreover, all-trans retinoic acid induced SOD and glutathione transferase activities, while it decreased MDA and reactive oxygen species in both healthy and varicocele sperm, suggesting that retinol enhances antioxidant enzyme activity [162]. Koda et al. [140] reported that all-trans retinoic acid supplementation (5 mg/kg per os) of adult ovariectomized rats significantly inhibits BPA (100 mg/kg)-induced uterine weight increase, inhibiting the estrogenic activity of BPA. Interestingly, pomegranate juice (containing vitamin A) prevents BPA-induced structural changes in rat epididymis, increasing the number of caudal sperm and decreasing sperm abnormalities [163]. On the other hand, Shmarakov et al. [164] recently reported that although hepatic retinoids are required for BPA biotransformation and its excretion, retinoid intake (3000 IU of retinyl acetate, by gavage) can enhance the noxious effects of BPA intoxication (50 mg/kg daily), increasing non-mitochondrial ROS production.

### 5.2. Natural Extracts and Products

Among other prophylactic or therapeutic approaches, the use of natural substances such as herbals and phytochemicals should be considered. 

Gallic acid, a polyhydroxyphenolic compound, is mainly found in vegetables, fruits (i.e., pineapple, banana, grapes, gallnuts, apple, and lemons), and plants, as free or part of hydrolyzable tannins. The antioxidant properties of this compound, linked to sequestering metal ions and ROS scavenging, has been demonstrated (reviewed in Badhani et al. [165]). Very recently, the role of gallic acid (20 mg/kg in corn oil) has been determined in testicular oxidative stress caused by chronic BPA exposure (10 mg/kg) in adult rats [141]. These authors showed that gallic acid increased antioxidant enzymes, reduced testicular lipid peroxidation (MDA, MPO), and normalized the gonadosomatic index (used to determine the sexual maturity in correlation to testes development), all parameters that were altered by BPA.

Among fungus of traditional Chinese medicine, *Cordyceps militaris*, used for the treatment of impotence, seminal emission, and infertility, displays a wide range of biological activities, including antioxidant effect due to polysaccharides and cordycepin content (reviewed in Yue et al. [166]). It has been investigated the protective role of *Cordyceps militaris* extract (200–800 mg/kg) in counteracting the reproductive damage associated with BPA toxicity (200 mg/kg) in rats [142]. The dose-related effect of this natural product resulted in the increase of sperm count and motility compromised by BPA, as well as in reducing the oxidative stress and the related testicular histopathological changes and dysfunction. These effects were related to the enhancement of antioxidant enzyme activity and GSH levels [142].

*Cuscuta chinensis* flavonoids (CCF), the main components of Chinese herbal medicine *C. chinensis*, show antioxidant and anti-abortive activities (reviewed in Donnapee et al. [167]). Recently, it has been reported that these flavonoids (40 mg/kg) can reduce the apoptosis of testicular cells induced by BPA (5 mg/kg) [143]. This effect is directly related to oxidative damage in male mice offspring (F1), after contextual maternal exposure to BPA and flavonoids. Specifically, CCF blocked the transcription and translation of apoptotic proteins (i.e., caspase 7 and 9) in the testes of F1 mice collected at postnatal days 21 and 56, reducing structural damage and ameliorating their fertility [143]. 

Very recently, the protective effects of hydroethanolic *Murraya koenigii* extract of leaves (200 mg/kg) in orally BPA (1 mg/kg) treated mice were determined [144]. In testis, BPA induced a significant decrease in sperm parameters, germ cell number, along with increased LPO, ROS, and apoptotic proteins, and all these alterations were recovered by the *Murraya koenigii* extract.

### 5.3. Melatonin

Melatonin is a neuro-hormone derived from tryptophan mainly released from the pineal gland that acts on the hypothalamic–pituitary–gonadal axis and is involved in different homeostatic activities, including reproduction managing, circadian rhythms and redox balance maintenance [168]. Melatonin has gained considerable attention as an antioxidant, as shown by several experimental and clinical studies highlighting its safety [169]. Indeed, melatonin acts as a scavenger of ROS and nitrogen species, including hydroxyl radical, peroxynitrite anions, and H_2_O_2_, as well as an inducer of antioxidant enzyme transcription. The chemical structure of melatonin allows its crossing through the biological membrane, reaching cytosol, mitochondria, and nucleus, preventing DNA damage and promoting signaling function [170]. A previous study evidenced the protective effect of melatonin (10 mg/kg, i.p.) on mitochondrial toxicity induced by short-term exposure to BPA (10 mg/kg) in testicular mitochondria of adult mice [145]. It has also been reported that melatonin (10 mg/kg, i.p.) was able to counteract BPA (50 mg/kg)-induced oxidative damage and apoptosis in rat testes and the alteration of epididymal sperm after a long-term exposure [21]. Melatonin integrated the capability to manage cellular redox balance to the anti-apoptotic effect on rat germ. Indeed, melatonin positively modulated glutathione, SOD, and catalase, as well as MDA levels and H_2_O_2_ yield in the testes and sperm and reduced apoptosis by inducing the anti-apoptotic and redox-sensitive protein Bcl-2 [21]. 

The control of cellular redox balance by melatonin is also strictly associated with its protective activity on meiosis of spermatocytes and normal sperm quality and production, due to modulation of serum testosterone levels, that was evidenced in both animals exposed or not to BPA [21,171]. 

Since the efficacy of melatonin supplementation was demonstrated on DNA changes, it has been proven that the hormone (10 mg/kg) improved BPA potential genotoxicity (200 mg/kg), reducing the damaged DNA in male germ cells of adult rats via the suppression of oxidative stress [146]. In particular, the melatonin prophylactic effect resulted in a decrease of DNA migration within germ cells and of assembly of γH2AX (a marker of DNA double-strand breaks), as observed by immunostaining of spermatocytes. This effect was accompanied by a melatonin-induced reduction of thiobarbituric acid reactive substances levels and an increase of SOD activity in testicular cells [146]. 

Olukole et al. [172] reported the protective effect of melatonin on the prostate function, as well as on the adrenal function [173], following a 2-week oral BPA exposure in the adult male rats. Later, Olukole et al. [174] also explored the relevant role of melatonin (1 mg/kg) in protecting F1 adult rats against male reproductive toxicity, following maternal exposure to different doses of BPA. The antioxidant protective effect of melatonin on interstitial necrosis in testis (caused by BPA administration in utero) was related not only to the improvement of testicular SOD, GSH, and GPx activity but also to the limitation of germinal cell degeneration, recovering BPA-altered tubular and luminal diameter [173].

Very recently, Akarca-Dizakar et al. [175] confirmed the beneficial effect of melatonin (20 mg/kg) on the alteration of epididymal structure and sperm quality induced by chronic exposure to BPA (25 mg/kg).

BPA has also been reported to adversely affect the female reproductive system. Zhang et al. [112] determined the mechanisms of the detrimental effect of BPA (100 μg/kg bw per day for 7 days) on the oocyte quality (oocyte meiotic maturation and fertilization ability) in mice. The authors observed that oral administration of melatonin (30 mg/kg bw per day for 7 days) elevated the in vitro fertilization rate, restoring BPA-induced alteration of fertilization proteins and processes via reduction of ROS levels and inhibition of oocytes apoptosis.

Moreover, melatonin protects the uterus degeneration induced by BPA exposure during the neonatal period, as histologically and morphometrically evidenced [176].

### 5.4. Selenium

Much evidence have allocated an important role of selenium in improving spermatogenic oxidative damage and apoptosis and in maintaining male fertility upon BPA administration [105,147]. Accordingly, it has been previously demonstrated the capability of selenium in managing the critical balance between germ cell death and proliferation during apoptosis in spermatogenesis [177]. In mice exposed to BPA (150 mg/kg), Kaur et al. [147] found that selenium supplementation in the diet (0.5 mg/kg diet) limited ROS and lipid peroxidation in mouse testes and ameliorated histopathological changes, preserving basement membrane and reducing germ cell vacuolization. This effect is probably due to the interaction between selenium and selenoproteins, including GSH-Px, an enzyme with ROS scavenging properties.

A recent study confirmed the antioxidant activity of selenium (3 mg/kg) and nanoselenium (2 mg/kg) on reproductive toxicity induced by BPA (150 mg/kg) in male rats [105]. Nanoselenium is a nanoparticle of selenium characterized by higher bioavailability, surface activity, and lower toxicity than selenium [178]. The protective effect of both elements on BPA-induced testicular toxicity in rats has been demonstrated; DNA fragmentation damage, inflammation, the expression of specific genes involved in spermatogenesis, and oxidative stress biomarkers were improved by both elements [105]. 

### 5.5. Methyl Donors

Among other therapeutic approaches, the treatment of N-acetylcysteine was found to improve BPA-induced adverse metabolic effect and oxidative stress in rats [129]. Cysteine plays a role in the sulfation cycle, acting as a sulfur donor and as a methyl donor in the conversion of homocysteine (Hcy) to methionine. Hcy undergoes auto-oxidation, generating ROS, and contributing to the onset of reproductive disorders. At high concentrations in the serum pregnant women, Hcy causes the low birth weight of offspring, preterm birth, and intrauterine abortion [179,180]. Maternal dietary supplementation with methyl donors, at doses able to reduce Hcy level in plasma, improved offspring’s intestinal digestion and absorption dysfunction induced by BPA, which might be associated with DNA methylation [181].

Supplementation with vitamin B_9_, B_6_, and B_12_ and betaine reduces plasma Hcy levels and ameliorates reproductive performance, as demonstrated in sows [149,150]. In agreement with these findings, Mou et al. [148] have recently demonstrated that maternal methyl donor supplementation (3 g/kg betaine, 400 mg/kg choline, 150 μg/kg vitamin B_12_, and 15 mg/kg folic acid) in the diet during gestation effectively counteracted BPA-induced placenta oxidative stress and contributed to maintaining a healthy redox environment in both fetus and offspring pigs. Actually, a dietary methyl donor limited BPA detrimental effects on placental integrity and increased the antioxidant activity of enzymes (SOD, catalase, and GSH-Px) acting on placental trophoblastic cells and umbilical cord blood, which then reflected on redox balance maintenance in newborn piglets [148]. 

Interestingly, Zhuo et al. [182] studied the effects of maternal nutrition on muscle characteristics in offspring after prenatal BPA exposure and co-treatment with methyl donors. BPA increased lactate dehydrogenase (LDH) enzyme activity and mRNA levels in the thoracic muscle at birth and the finishing stage, and reduced methylation at the LDH promoter. The protective treatment with methyl donors results in persistent effects on pork quality meat, affecting glutathione and LDH expression via DNA methylation.

## 6. Conclusions

BPA has become a target of intense research based on its metabolic and endocrine interference and its association with human diseases such as obesity, diabetes, reproductive disorders, and cancer. This review provides a framework for understanding how the induction of oxidative stress may contribute to the pleiotropic reproductive adverse effects observed after BPA exposure. The numerous examined studies show that reproductive organs in males are more vulnerable to BPA exposure than those in females. Evidence suggests that males may be more susceptible for several reasons (i) gender differences in glutathione levels and their relevance for detoxification process (lesser glutathione availability in males); (ii) greater sulfate-based detoxification capacity in females; (iii) greater inflammatory response in male reproductive organs; (iv) reduced vulnerability to oxidative stress in female organs. 

Perinatal and developmental exposure to BPA can induce oxidative stress and lipid peroxidation; these effects are conserved across animal species and humans and can disrupt metabolic and endocrine physiological functions in the reproductive system. Often, oxidative stress appears as the primary abnormality in an organ; then, inflammation will develop and further exacerbate oxidative stress and vice versa.

Despite differences among doses, duration, model systems, and measured outcomes, growing and compelling evidence reports that wide variety of doses or concentrations of BPA (in vivo and in vitro or human exposure) promote ROS generation and alteration of redox balance, inducing mitochondrial dysfunction and the modulation of cell signaling pathways related to oxidative stress. The induction of oxidative stress by BPA can act in a dependent- or independent-manner to its endocrine and metabolic disrupting properties that may induce marked reproductive effects during prenatal, perinatal, and postnatal exposure or in adulthood. At the same time, the effect of BPA can be particularly severe when the exposure is associated with other risk factors, such as poor diet, metabolic impairment, and concurrent diseases. 

To date, different antioxidant approaches have been identified to counteract BPA-induced damage in male and female reproductive systems. These agents improve male and female fertility, reducing multiple oxidative stress markers, lipid peroxidation, or DNA damage and restoring the antioxidant defense. However, new experimental and clinical studies are warranted to establish the specific molecular mechanisms underlying the protective effect of these antioxidant substances on oxidative stress and inflammation responsible for systemic and organ-specific toxicity of this ubiquitous xenobiotic. Further studies on antioxidants are needed to better define their usefulness in several gender-dependent reproductive dysfunctions and to establish their optimal dosage and scheme of treatment for counteracting BPA toxicity.

## Figures and Tables

**Table 1 antioxidants-09-00405-t001:** Classification of the different types of antioxidant approaches against BPA toxicity on male and female reproductive systems.

Classification	Type	Dose	Notes
**Vitamins and co-factors**	**Glutathione + Vitamin E**	5 + 2 mM	This combination counteracts the compromised fertilization and early embryo development caused by 100 M BPA in vitro [90]
	**Vitamin C + Vitamin E**	50 + 50 mg/kg	The co-administration of these vitamins exert a protective effect in preventing apoptotic cell death in the ovaries of rats exposed to high dose of BPA [134]
	**Vitamin E**	4 mg/100 g bw	Improves male fertility, protecting testicular cells and epididymal sperm from the apoptosis induced by BPA exposure [135]
	**1,25-dihydroxyvitamin D_3_ (1,25D_3_)**	0.1 M	In vitro modulates the toxic effect of BPA (10 M) on oxidative stress and particularly on mitochondrial function and dynamics in ovarian granulosa cells. 1,25D_3_ prevents mitochondrial DNA deletion induced by BPA in female ovary [136]
	**Coenzyme Q10**	100 g/mL	Restores fertility, rescuing the reproduction toxicity induced by BPA (500 M) in the *Caenorhabditis elegans germline,* counteracting DNA damage and reducing oxidative stress through the scavenging of ROS and free radicals and restoring mitochondrial function [137]
	**α-Lipoic acid (LA)**	20 mg/kg 100 mg/kg	In male rats, LA reduces testicular and mitochondrial oxidative stress caused by chronic administration of BPA (10 mg/kg), normalizing the activity of key mitochondrial enzymes and increasing enzymatic and non-enzymatic antioxidants of mitochondria [138] In female rats, long-term treatment with LA, during BPA exposure (25 mg/kg) for 30 days, reduces ovarian oxidative damage, preventing lipid peroxidation and nitrosative stress, even more in combination with vitamin E (20 mg/kg, per os) [139]
	**Retinoic acid**	5 mg/kg	All-trans retinoic acid supplementation of adult ovariectomized rats inhibits the uterine weight increase induced by BPA (100 mg/kg) inhibiting estrogenic activity of BPA [140]
**Natural products**	**Gallic acid**	20 mg/kg	Counteracts testicular oxidative stress caused by chronic BPA exposure (10 mg/kg) in adult rats, increasing antioxidant enzymes and reducing the markers of lipid peroxidation, i.e., MDA and MPO [141]
	***Cordyceps militaris* extract**	200–800 mg/kg	The dose-related effect of this extract against BPA toxicity (200 mg/kg) on reproductive system in rats results in the reduction of oxidative stress and testicular histopathological changes, and in the enhancement of antioxidant enzyme activity and GSH levels [142]
	***Cuscuta chinensis* flavonoids (CCF)**	40 mg/kg	The protective effect of CCF on the apoptosis of testicular cells induced by BPA (5 mg/kg) is directly related to oxidative damage, in male mice offspring (F1), after contextual maternal exposure to BPA and flavonoids. CCF blocked the transcription and translation of apoptotic proteins (i.e., caspase 7 and 9) in the testes of F1 mice [143]
	***Murraya koenigii* extract**	200 mg/kg	This hydroethanolic leaves extract improves sperm parameters and reduces LPO, ROS and apoptotic proteins in the testes of BPA (1 mg/kg) treated mice [144]
**Hormones**	**Melatonin**	10 mg/kg	Reduces mitochondrial toxicity induced by short-term exposure to BPA (10 mg/kg) in testes of adult mice [145] Counteracts BPA (50 mg/kg)-induced oxidative damage and apoptosis in rat testes after long-term exposure. Improves glutathione, SOD and catalase, MDA levels and H_2_O_2_ yield, as well as induces the anti-apoptotic and redox-sensitive protein Bcl-2 [21] Improves BPA potential genotoxicity (200 mg/kg), reducing the damaged DNA in male germ cells of adult rats via the suppression of oxidative stress. In particular, it limits DNA migration within germ cells and, simultaneously, reduces thiobarbituric acid reactive substances levels and increases SOD activity in testicular cells [146]
**Trace elements**	**Selenium** **Selenium/** **Nanoselenium**	0.5 mg/kg 3 mg/kg 2 mg/kg	The supplementation in the diet results in the reduction of ROS and lipid peroxidation and histopathological changes in testes of mice exposed to BPA (150 mg/kg). This effect is probably due to the interaction between selenium and selenoproteins including GSH-Px, an enzyme with ROS scavenging properties [147] The protective effect of selenium and nanoselenium on BPA (150 mg/kg) induced testicular toxicity in rats results in the improvement of DNA damage, inflammation, the expression of specific genes involved in spermatogenesis and oxidative stress biomarkers [105]
**Methyl donors**	**Betaine** **Choline** **Vitamin B_12_** **Folic acid**	3 g/kg 400 mg/kg 150 g/kg 15 mg/kg	Maternal dietary supplementation during gestation counteracts placental oxidative stress induced by BPA (50 mg/kg diet), increasing antioxidant activity of SOD, catalase, and GPx, and then reflects on redox balance maintenance in newborn piglets [148]
	**Vitamin B_12_ + Folic acid** **Betaine**	15 + 15 mg/kg 20 + 150 g/kg 3 g/kg	Supplementation with vitamin B_12_+folic acid [149] or with betaine [150] in the diet reduces plasma Hcy levels and ameliorates reproductive performance, as demonstrated in sows during gestation

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
