# Peer review of "Oxidative Stress and BPA Toxicity: An Antioxidant Approach for Male and Female Reproductive Dysfunction"

_antioxidants, 2020, doi:10.3390/antiox9050405_

Round 1

Reviewer 1 Report

This is an interesting review article on the effects of BPA on reproductive physiology, including its action mechanism, and describes five classes of antioxidants that have been suggested to counteract the detrimental effects of exposure to BPA. Overall, the Manuscript is well written, contains relevant information and falls into the scope of Antioxidants. I just have a couple of Major Concerns.

MAJOR CONCERNS

My main concern is with the type of review article, since the authors state in the title that it is a systematic review, but PRISMA guidelines do not seem to have been fulfilled and the study does not appear to have been registered at PROSPERO. If authors really performed a systematic rather than a narrative review, they should then include a M&M section indicating how inclusion of articles was performed (criteria) and which the PROSPERO registration number is. If they did not conduct a systematic review, then please remove the words ‘a systematic review’ from the title

The second point regards to the lack of an integrated narrative throughout the Manuscript, especially in the case of Sections 1, 2 and 6. There too many paragraphs that, in my opinion, could be merged, which would make the paper easier to follow. Moreover, some information is repeated in separate subsections, so that I suggest to revise the Manuscript thoroughly and be as succinct as possible, avoiding reiteration

Finally, I think that some passages are supported by previous review articles. However, I strongly recommend authors to use critical, original research articles rather than reviews.

SPECIFIC COMMENTS

Pg1 L17 not clear why authors conclude the sentence by stating that exposure to BPA results chronic. Do they mean that ‘exposure is prolonged’?

Pg1 L40 ‘environments’

Pg2 L38 ‘Mounting evidence shows’

Pg3 L22 ‘These authors assumed’

Pg4 LL1-8 Please use critical literature rather than a review

Pg4 L12 ‘mitochondrial mass’? Not clear what authors intended to mean

Pg4 LL17-26 I think that these two paragraphs could be merged

Pg4 L27 ‘suggests’

Pg4 LL35-37 Please revise this sentence

Pg4 LL41-42 It should be explained how these defects are caused, and some figures to understand the actual impact of BPA exposure

Pg4 L47 name of species in italics

Pg5 L19 ‘DNA may be…’

Pg5 L28 and L30 Citations required

Pg5 L35 ‘heritable. Methylation…’

Pg5 L37 Reference missing

Pg5 LL34-42 Consider merging these two paragraphs

Pg5 L47 ‘leading to the disruption…’

Pg5 LL47-50 Please revise this sentence

Pg5 L4 ‘determined’ or ‘investigated’ rather than ‘assessed’?

Pg5 L9 ensure that all abbreviations are defined when used for the first time

Pg5 LL9-11 In my opinion, review articles should not contain unpublished results. Please remove if this paper has not already been published

Pg5 L30 Avoid starting a sentence with an abbreviation

Pg5 L48 This citation does not follow the Journal guidelines

Pg6 LL8-22 Link to reproductive physiology unclear

Pg6 L32 ‘levels’

Pg6 L33 ‘motility’ rather than ‘mobility’

Pg6 L39 ‘their fertilizing ability’

Pg6 LL42-49 Please provide stronger link to reproductive physiology

Pg8 L25 SIRT1 should be written in italics if referred to the gene

Pg8 L29 ‘it is demonstrated’

Pg9 L29 Remove ‘leaving the mother’

Pg10 L2 ‘also involves’

Pg10 L5 ‘the capability of BPA to induce’

Table 1: References should be given in numbers, as per the Journal guidelines

Pg15 L19 ‘found to be reduced’

Pg15 L19 ‘sequestering metal ions…’

Pg16 L1 ‘These authors showed’

Pg17 L5 ‘also explored’

Pg17 L44 ‘At high concentrations in the serum of pregnant women,’

Pg18 L42 ‘underlying’?

Reviewer 2 Report

It is a very good review focusing on the OS/BPA and reproduction.

Comments:The topic chosen is very broad. The authors may consider split the review article into two (OS/BPA-males; OS/BPA-female). If the authors like to keep both genders, then they have to detail many aspects of the manuscript.

Page 4, lines 28-36: The authors may consider rewrite this paragraph.

2.. Oxidative stress and BPA: This is a very important/key section for this manuscript. This section is not written well. The authors started this section stating that BPA induce OS; BPA- inflammation; BPA- cancer; BPA- pancreatic dysfunction ;  BPA- neuron degeneration ;  BPA- autism etc. The reviewer agree with teh authors that BPA induce OS in different cell types/diseases/systems. However the authors need to consider the different systems (Cancer signaling is different from toxicology vs neuroendocrine mechanisms). The authors need to focus only on the reproductive system and the associated OS mechanisms.
3. Mechanisms linking BPA-induced oxidative stress to reproductive system
Under this section also the authors should restrict to the information on the (in vitro/ in vivo) reproductive systems not on the cancer/tumor cell lines or other systems.

Section 4 and 5 needs to be elaborated.

Reviewer 3 Report

In the present review the authors aimed at examining how the induction of oxidative stress can be considered the ‘fil rouge’ of BPA's toxic mechanisms on reproduction, and they focused their attention to the protective effects of different classes of antioxidants.

The paper is interesting, but there are some points inside the manuscript that should be clarified.

Please write the word “non persistent” in the same way all over the paper (nonpersistent or non-persistent).

In paragraph 2 authors should also talk about nitric oxide and peroxynitrite as nitrogen reactive species induced by BPA.

Make sure that all acronims are defined the first time they appear (page 6 line 9: SOD, GST, and GSH‑Px)

Page 7 line 12: plase consider to rephrase the sentence because leptin is able to modulate the production of inflammatory mediators

Page 7 line 51: Wang and Cheng

Page 8 line 33: please insert a reference

Page 15: it is not clear, at least to me, why authors only describe Lipoic Acid and no description for other compounds (Vitamins and cofactors). The same in the following pages.

Authors should mention also retinoic acid

The manuscript needs careful editing to improve the flow of information, some parts are very hard to understand and to follow.

Round 2

Reviewer 1 Report

The authors have addressed my previous concerns. I have no further comments. A suggest, however, removing the words 'a narrative review' from the title. My previous concern was that the title stated that the Manuscript was a systematic review, but this was not the case. Therefore, rather than indicating that the review is narrative, my suggestion is for not stating that it is systematic.

Reviewer 2 Report

Page 4, lines 7-9: The authors stated that "... Moreover, in females, mitochondria have higher levels of reducedglutathione than those of males and the oxidative damage to mitochondrial DNA is lower in females than in male ...". This interesting section may be detailed.
Page 5, line 32-33: (our unpublished data, presented as oral communication
33 to the international congress of British Pharmacological Society 2019, Edinburgh, UK). - this may be rephrased as 'unpublished results'
Page 5: The unpublished results may be detailed along with the epigenetic changes.
